# Open Source Assessment of Deep Learning Visual Object Detection

**DOI:** 10.3390/s22124575

**Published:** 2022-06-17

**Authors:** Sergio Paniego, Vinay Sharma, José María Cañas

**Affiliations:** Departamento de Sistemas Telemáticos y Computación (GSyC), Universidad Rey Juan Carlos, 28942 Fuenlabrada, Madrid, Spain; sergio.paniego@urjc.es (S.P.); vinay.sharma@gsyc.urjc.es (V.S.)

**Keywords:** object detection, open-source, software tools, model evaluation

## Abstract

This paper introduces Detection Metrics, an open-source scientific software for the assessment of deep learning neural network models for visual object detection. This software provides objective performance metrics such as mean average precision and mean inference time. The most relevant international object detection datasets are supported along with the most widely used deep learning frameworks. Different network models, even those built from different frameworks, can be fairly compared in this way. This is very useful when developing deep learning applications or research. A set of tools is provided to manage and work with different datasets and models, including visualization and conversion into several common formats. Detection Metrics may also be used in automatic batch processing for large experimental tests, saving researchers time, and new domain-specific datasets can be easily created from videos or webcams. It is open-source, can be audited, extended, and adapted to particular requirements. It has been experimentally validated. The performance of the most relevant state-of-the-art neural models for object detection has been experimentally compared. In addition, it has been used in several research projects, guiding in selecting the most suitable network model architectures and training procedures. The performance of the different models and training alternatives can be easily measured, even on large datasets.

## 1. Introduction

The detection of objects on images and videos is a foundational area of research in computer vision consisting of location and classification of class instances of objects on images. It has gained enormous popularity in the last decade thanks to some milestone advancements, which can be observed in the number of high-quality reviews published recently on this topic [1,2,3,4,5,6,7]. The number of real-life applications that implement it also exposes its importance. It is used on autonomous driving applications [8] that need to understand the nature of the surrounding objects or in-camera filters and effects based on objects, for example, faces [9]. Detection of objects in retail scenarios [10] can be another application area or even traffic monitoring, as investigated in the Experimental Results section of the present paper.

This progress has occurred due to a sequence of favorable factors and remarkable events. For example, the great variety of massive open access quality datasets focused on object detection and the development of deep learning architectures, especially convolutional neural networks, with an impressive performance that have become notorious over time. The availability of high computational power provided by GPUs is another factor that has influenced this great scientific progress in the last few years. In the datasets side, COCO [11], Imagenet [12], Pascal VOC [13], Princeton [14], Spinello [15] and Open Images Dataset [16] are some of the most commonly adopted for this topic. These object detection datasets and deep neural network models are constantly improved, reaching better performance results over time and proving the current importance of this problem in research and industry. Nevertheless, the object detection problem is not yet completely solved, exhibiting flaws in situations such as small object detection [17] or traffic sign detection [18].

The object detection datasets are typically generated using software tools for annotation [19], labeling the ground truth class and bounding box for each interesting object instance that the model will learn on every image on the dataset.

Learning from a dataset with only a few annotated examples for each class, known as few-shot learning, is also a current topic of interest in object detection [20,21]. It is partly based on the evidence that humans only need a few examples or even one to learn to classify an object and distinguish it. Other predominant areas of computer vision that are closely related to object detection and whose research usually improve in parallel are image classification [22] and image segmentation [23,24,25].

Developing a deep neural network model for object detection involves several steps. One of the steps is to select the deep learning framework to code the actual network architecture. The available options comprise TensorFlow [26], Caffe [27], Darknet [28], Keras [29] or PyTorch [30]. Another step could be selecting one of the commonly used architectural techniques for constructing the network. Some of them are Faster Regional-CNN [31], Single Shot MultiBox Detector [32] and You Only Look Once (YOLO) [33].

Additionally, the typical workflow involves trial and error, changing some hyperparameters in the network architecture and retraining it several times to improve its performance results. This situation brings the problem of objectively compare the performance of those different trained neural network models with the same conditions so a developer can quickly understand which model is better for a use case. The main contribution of this paper, the Detection Metrics software tool, aims to help researchers in the testing of object detection neural networks by providing objective performance metrics (on massive datasets) so the user can easily compare different networks or training parameters and see which one performs better. Its application is not focused on any particular task inside object detection so it can be applied to a variety of projects. Several different tools form Detection Metrics, providing each one with some helpful features to objectively compare different neural network models for object detection. The Detection Metrics toolkit is open-source and it is accessible via https://github.com/JdeRobot/DetectionMetrics, accessed on 27 April 2022.

## 2. Related Work

In this section, we conduct a review of relevant state-of-the-art related work in object detection using deep learning.

### 2.1. Datasets

In recent years, the research community has introduced many free-access image datasets for object detection. Some of them include competitions that have helped boost the advancements in object detection and computer vision in general. Some of them, supported in Detection Metrics, are the following:ImageNet [12]: ImageNet is the largest public collection of images, containing 14,197,122 samples, where 1,034,908 images have been annotated with bounding boxes, ideal for training and evaluating object detection models.Pascal VOC [13]: Pascal VOC’s 2012 release contains 11,530 images in training and validation datasets, spanning 20 classes. It encloses a total of 27,450 bounding box annotated objects.Spinello dataset [15]: Spinello is a dataset consisting of 3000+ RGB-D images captured using a Microsoft Kinect containing people. It focuses on person detection and tracking in 3D space.Princeton RGB dataset [14]: This dataset contains 100 RGB-D videos of high diversity focused on the design and comparison of tracking algorithms. It is similar to Spinello and also uses a depth sensor to capture images.Common Objects in Context (COCO) [11]: COCO is designed for both object detection and segmentation. It contains around 330,000 images of which 200,000 are labeled, containing 1.5 million object instances in total.Open Images Dataset [16]: Open Images Dataset V6 is the largest existing dataset with object location annotations, containing 1.6 M images with 16 M bounding boxes for 600 object classes.

Even though they are not natively supported in Detection Metrics, there are additional image datasets focused on more specific tasks, such as traffic signs [34], small object detection [35] or human detection in crowd scenarios [36]. Since Behavior Metrics is open-source, support for new datasets can be implemented easily.

### 2.2. Frameworks

Object detection deep learning models are implemented using different deep learning frameworks [37] instead of coding everything from scratch. These frameworks have common pre-built code components, ready to use and optimized for performance, making it easier for a programmer to build a model. These frameworks are often open-source, are developed in the most common programming languages such as Python or C++ and some have additional support from private companies. For example, Google develops Tensorflow [26] and Keras [29]. Both frameworks provide instruments to create networks based on tensors, with Tensorflow providing a deeper control for the user and Keras acting as a wrapper for Tensorflow’s functionality. PyTorch [30] is an open-source framework developed by Meta that has gained strong popularity among the research community in the last few years. Additionally, Caffe [27] and Darknet [28] are other open-source frameworks.

### 2.3. Performance Metrics

Various performance metrics are frequent in object detection benchmarks. They help understand the performance of the different architectures and decide the most suitable option for a concrete scenario. A model with a slightly worse performance result on some metrics but, for example, with a better mean inference time, could be more useful in certain conditions than others with different results. Some of the most common metrics in object detection that are also present in Behavior Metrics for evaluation are:Average Precision (AP): Fraction of the total amount of correct predictions. Ranges from 0 to 1.
(1)$$Precision=\frac{True\;positives}{True\;positives+False\;positives}$$Average Recall (AR): Fraction of the total amount of predictions that are detected. Ranges from 0 to 1.
(2)$$Recall=\frac{True\;positives}{True\;positives+False\;negatives}$$Mean Average Precision (mAP) and mean Average Recall (mAR): We usually consider a range from 0.5 to 0.95 for Intersection over Union (IoU). The IoU metric compares the ground truth bounding box with the detected ground truth and retrieves a value between 0 and 1, indicating how close the detected bounding box is to the ground truth. The higher the value of IoU, the closer to the ground truth bounding box. In the present work, mAP and mAR with IoU values from 0.5 to 0.95 is calculated (IoU = 0.5:0.95) for the dataset classes.
(3)$$IoU=\frac{Area\;of\;overlap}{Area\;of\;union}$$Mean inference time: Average time spent generating predictions. In milliseconds, the importance of these metrics depends on the scenario where the model is applied. This metric could be relevant in environments that imply generating fast and precise answers.

### 2.4. Network Models

There are two main groups in the classification of object detection network architectures: Region Proposal-Based Framework and Regression/Classification-Based Framework, as classified in [3]. In the first one, a chain of correlated steps is conducted. These differentiated steps usually lead to a bottleneck in real-time. In the second group, the techniques are based on regression and only involve one global step, improving the computation time.

Other methods have introduced transformer architectures [38] that base their workflow on attention mechanisms and have obtained remarkable experimental results with a promising future [39,40].

We briefly introduce an explanation of three of the most successful network architectures, including Faster Regional-CNN (Faster R-CNN) in the Region Proposal-Based Framework group and Single Shot MultiBox Detector (SSD) and You Only Look Once (YOLO) in the Regression/Classification-Based Framework group.

#### 2.4.1. Faster Regional-CNN

Faster R-CNN [31] is a multi-component detector comprising a Region Proposal Network (RPN) that first generates highly probable regions. Later layers classify these region proposals and, in the last step, a bounding box regressor reduces the localization error of the predicted bounding boxes. Faster R-CNN is an improvement upon R-CNN and Fast R-CNN to make it more real-time and robust. Faster R-CNN also generates much fewer region proposals compared to R-CNN and Fast-RCNN leading to reduced detection time while simultaneously maintaining detection accuracy.

Stages in Faster R-CNN:Anchor generation: Anchors are regions that may contain an object. So, anchor generation must be as thorough as possible because if a particular region is mixed then there is no way that it would be detected in the succeeding layers. These anchors are later refined using a bounding box regressor to reduce the localization error to better localize objects. Anchor Generation uses sophisticated algorithms to cover the whole image, such as selective search, which is later fed into the Region Proposal Network (RPN).Region Proposal Network: The job of this component of the network is to generate regions with a high probability of containing objects. It takes anchors as input and produces highly probable regions. Again, if a region containing an object is not proposed then there is no way that the component would detect it in the following layers. Moreover, the number of regions should be as low as possible to reduce detection time and as thorough as possible to reduce false negatives.Classifier and bounding box regressor: The final component of the network classifies proposals from RPN into an object class or background (i.e., negative or no object present). Classification occurs first and then its results are better localized to reduce the localization error or to accurately place the bounding box on the classified object. This component regresses four parameters, namely *x, y, w,* and *h*, where *x* and *y* are the top-left coordinates and *w* and *h* are the width and height of the bounding box.

#### 2.4.2. Single Shot MultiBox Detector

SSD [32] (SSD) proposes a more unified approach toward object detection compared to Faster R-CNN in which detections are generated in a single forward propagation of a unified network. This approach uses different techniques to propose regions. The whole working pipeline, including region proposal, classification, and bounding box regressor (to reduce localization loss), is part of a single unified network, which significantly increases the prediction speed.

Grids which can be 8 × 8 or 4 × 4 divide the given image into grids. For each box present in the grid, SSD predicts the offsets for the bounding box present in that grid and the confidence score (probability of each class in the particular region) for all object categories. Then each box is matched to a single class, and the final results are used to compute loss, including both classification and localization loss.

#### 2.4.3. You Only Look Once (YOLO)

The idea of SSD’s unified detection inspired this network architecture. It also uses a similar system to the one introduced by SSD to propose regions for further classification and regression.

Similar to SSD, YOLO [33] also divides the input image in a grid, and the grid size is variable (i.e., depends on the dataset and the type of problem it is being used for). Assuming a grid size of *S* × *S*, for each grid cell *B* bounding boxes are generated where *B* is also a variable and depends on the dataset and the type of problem it is being used for. For instance, B=2 for Pascal VOC dataset. After generation, these bounding boxes are sent for classification and regression, to output final bounding box predictions. Additionally, YOLO also has a very creative loss function, which takes care of both the classification and localization error, i.e., a single combined loss function is used to minimize both classification and localization error.

The prime selling point of YOLO was real-time detection which was made possible by using its unified network proposal. The accuracy for YOLO is great, in certain cases lower than other network models but the speed is high, making its use in real-time scenarios possible.

From its initial release, several improvements have been applied to YOLO until the last versions available, YOLOv4 [41] and YOLOv5 [42]. For example, in YOLOv3 [43], bounding boxes using dimension clusters as anchor boxes are included, proposed in [44] or a hybrid feature extractor approach between YOLOv2 and residual networks. YOLOv4 and YOLOv5 follow the same idea of incremental improvements taking different ideas that have been proven to work and making YOLO more efficient.

## 3. Detection Metrics Tool Kit

Detection Metrics, the main contribution of this project, is a multi-platform command-line and graphical software application that provides several tools for comparing deep learning architectures for object detection images. Its GUI is based on *Qt* framework (see Figure 1) and written in *C++*. The software application is supported by the most common operating systems providing it as a Docker image. Thanks to this technology, the functionality remains the same independently from the platform. It accepts neural network architectures trained using different frameworks: Tensorflow 2, Keras 2, PyTorch 1, Caffe 2 and Darknet (provided using the YOLO-OpenCV 4.2 module). Additionally, Detection Metrics provides support for a wide variety of dataset formats, having COCO, ImageNet, Pascal VOC, Princeton RGB, Spinello, and Open Images datasets supported.

The application supports two primary use cases: batch evaluation (headless mode) and live detection. The following sections cover these use cases in detail. A third supported use case is using Detection Metrics as a ROS Node inside a distributed application. The nodes are the building blocks of the applications in the Robot Operating System (ROS) approach. In this workflow, Detection Metrics acts as an executable node integrated in the distributed robotics application. It is a node that performs live detections, shares them with other ROS nodes, captures datasets, and stores metrics. For example, a ROS based application for a robot endowed with an onboard camera may use Detection Metrics this way for experimentally comparing object detection deep learning models.

### 3.1. Global Architecture and Workflows

The simplified architecture of the application can be illustrated as a black box (see Figure 2). It usually receives a batch of datasets and a group of deep learning trained models and it outputs objective metrics for the experiments generating predictions using the deep learning models over the datasets provided. This workflow is the headless evaluation.

Inside the black box, the application uses several tools that can also be used independently, especially when using the graphical part of the application. This use case allows the researcher to run several trained models over a batch of datasets easily at the same time, comparing their experimental performance and obtaining an idea of what is the best model for the problem that they are trying to solve.

Going into the detail of the architecture, there are six differentiated building blocks that integrate the tool set (see Figure 3). They are the Viewer, Detector, Evaluator, Deployer, Labeling and Converter. These six parts can be combined in three divided sections. Two of them are the main workflows or modules.

The second use case is live detection visualization. The main difference with the headless evaluation is that the sources for the live detection can be videos or live streams, for example, cameras. They generate detection online predictions and these predictions can be saved or even modified with the Labeling functionality. Finally, the Converter remains disconnected from the rest of the modules and common pipelines. Its application is the conversion between datasets formats.

Viewer is a tool used to display annotated datasets that is not part of the main workflows. When evaluating using the GUI, this means that while running, the user can view the detections the model is generating and compare them with the ground truth because both images and annotations are displayed. This tool supports several dataset implementations: COCO, Imagenet, Pascal VOC, Princeton RGB, Spinello and Open Images dataset. It also supports displaying and labeling depth images (for the datasets that give support to this feature) by converting them into a human-readable depth map.

Images are displayed one by one, showing the image with its corresponding detected objects with a bounding box and a tag label naming the class group it belongs to. The bounding box and label have different colors depending on their detected prediction.

When used separately from the evaluation, it provides slightly different functionality. Given a set of images and annotations, it displays them one by one. Additionally, the final annotated images that Viewer displays can be further filtered based on some specific classes (i.e., only particular classes will be labeled and only images containing those specific classes are displayed). This option can be interesting when looking for images that contain objects belonging to a specific class.

### 3.2. Headless Evaluation

The headless evaluation is one of the main use cases of the application. This mode is accessible directly via command-line. A researcher can determine a set of experiments that will run independently and unattended (fully autonomous), retrieving the final experiment report with the objective metrics that will help detect flaws and advantages for each model in each scenario.

This mode, as explained above, receives a batch of datasets and deep learning models, runs the inferences for each model over each dataset, and outputs objective metrics. Inside this process, three Detection Metrics tools are involved: Viewer, Detector and Evaluator. When working detached as headless, these three tools work together as one but they are additionally available separately when using the graphical user interface (GUI) application.

#### 3.2.1. Detection Generation

Detector tool is responsible for generating a new annotated dataset with predicted labels given a neural network trained model and a dataset. The generated dataset contains the images with the detected objects, their position in the image, and probabilities for the predictions. Different inference frameworks are supported: TensorFlow, Keras, Darknet, Caffe and PyTorch. When this tool is run, it also communicates with the Viewer to show the detections with the ground truth, giving an intuition of the performance visually.

In order to provide the different frameworks support, Detector has interfaces for each of them, connecting the actual deep learning framework to the tool in an agnostic way that prevents the user from facing any complexity. Thanks to the modularity of the Detector tool and the fact that the project is open-source, when new deep learning frameworks are created, they can be added without modifying the inner structure of Detector. In addition, the engineer or scientist saves evaluation time since they create the experiment description structure and it runs autonomously without explicitly consider the differences of the underlying frameworks or dataset structures and only focuses on the experimental results.

The console provides log information about the execution. This information shows for each image the predicted detected objects classes with their probabilities sorted. Additionally, information about the mean amount of time spent inferring the images is shown, a metric that is usually evaluated when the model is supposed to run in a live environment, instead of a situation where the inference time is not critical.

#### 3.2.2. Evaluation of Detections with Objective Metrics

Evaluator can evaluate two annotated datasets with the same dataset format on a fully autonomous basis considering one as the ground truth and the other as the generated detections dataset. Evaluator supports mAP and mAR metrics as described in the related work section of this paper or more detailed in COCO dataset paper [11]. It outputs mAP and mAR performance metrics for each class and for a range of IoU thresholds.

Every object detection in an image will be evaluated, comparing the detection in both datasets. Since the evaluation procedure in the application is written in C++, it provides faster performance than the original COCO toolbox written in Python. This procedure is done for every image in the dataset, loading, comparing and then releasing the resources, making a fair comparison.

When running in headless mode, the set of experiments is evaluated after the two previous steps and then creates a report in *csv* format with the experimental information.

Using the GUI, Evaluator can also be used independently, providing additional features. The evaluation can be further filtered by a specific object class from the detected dataset, so only the classes selected will be considered during the evaluation. There are two types of IoU available in Evaluator: bounding boxes and masks. Additionally, the different person classes available in some of the dataset class names can be merged into just one person class that contains all the different ones.

### 3.3. Live Detection Visualization

The second main use case is live detection visualization. One of the main differences between this use and the headless is the input source. For the Deployer, the main tool used for this use case, the input can be a video file or even a stream of images coming from a video camera—for example, a webcam. It receives the input and generates predictions in real-time on the images it receives, using the provided deep learning trained model.

Once the desired network model and framework for inferring have been chosen, the tool displays a video player that plays the input while displaying the objects detected with their class names in real-time. If the input is a video file, two video players are displayed, one of them with the raw video and the other one with the video and detected objects, similarly to Detector. This video player offers the flexibility to pause and play it again or go back and forward in the playback frame by frame. Another feature provided by Deployer is the confidence threshold (minimum value to consider a detection) that can be adjusted to different values to show the differences in the inferences in real-time. This will affect the real-time detections in the video since if the threshold is set to a high value the number of objects that will be found in a frame will probably be lower and the other way around, while the video plays, the console outputs the detected objects for each frame of the video with the percentage of confidence and the time spent on inferring.

The predicted labels can be saved to an output file if needed, setting an output folder, which for example can be used to create new image datasets with annotations from a video record or webcam output.

#### Labels Correction on Demand

Deployer provides the user with different tools related to labeling a dataset. The idea of generating human-annotated datasets has previously been described [19] and it is useful to complement the deep learning approach and its possible errors. This functionality is provided in the video player created when using Deployer.

The first feature is the possibility of adjusting the bounding boxes generated. The user can adjust the size and position of a certain detection bounding box stopping the video when the error is found and then adjusting the distribution of the box to the object.The second feature is changing the class name for every detected object. This means that a user can select a detected bounding box in the video image and change the class name in real-time to one of the class names provided or to a completely different one, also having the chance of adjusting the probability of the selection.The third feature is related to the previous ones and is adding new detections. The user can draw a new bounding box in a stopped frame and then give this a class name and probability.

This workflow can be interesting for generating completely new datasets for research of industrial purposes, creating first the proposals that some object detection model predicts and then adjusting the result by modifying the predicted bounding boxes and predicted classes or adding new predictions that are relevant for a specific task.

### 3.4. Dataset Converter

The dataset formats are usually specific for a certain implementation, so the purpose of this tool is to convert a dataset format into another format. This tool receives as input a dataset with the objects class names that are supported by it and the type of dataset format it implements. It needs the type of dataset as input to create a reader, a tool that understands the format for a specific dataset. The format implementation of the wanted dataset to be converted is also needed, so Detection Metrics creates a writer, another tool that knows how to write on a specific dataset format. Converter also gives the opportunity of filtering by object classes if it is provided with a set of class names, so a user can select which object classes to consider in the output dataset or even map the object classes to writer classes in the output dataset. This means that in the case that the object class names in the input and output dataset are different, the application tries to map from the input class names to the output ones, considering the common class name connection between the common datasets and also considering synonyms.

The converted dataset can be split into test and train parts. To do so, a training ratio is provided to the tool and it divides the dataset into two separated parts. This option can be useful to create divisions of the converted dataset.

After the conversion is completed, Viewer functionality can be used to display the converted dataset and make sure the process is completed successfully or it also can be used with the different tools provided by Detection Metrics.

## 4. Experimental Results and Discussion

This section describes two experiments conducted using Detection Metrics software. In the first experiment, we compare the performance of the most well-known state-of-the-art detection networks and validate the published results from the original network authors. In the second experiment, we demonstrate its usage in a real research application and how it has been used in the development process. It allows the iterative improvement of the networks providing objective feedback about their performance with real data. Both experiments validate the broad range of applicability of this scientific software.

### 4.1. Comparison of State-of-the-Art Detection Networks

In this experiment, four different pre-trained object detection networks are evaluated using Detection Metrics. The goal is to compare the results obtained by the toolkit with those published by the original authors. The selected networks include several popular object detection methods [3]: SSD, Faster RCNN and YOLOv3. In the process, the Headless evaluation mode of Detection Metrics (Figure 4) was used. The measured performance metrics are compared among them and also with those published by the authors.

The evaluation dataset is COCO minival, a small subset of COCO’s validation set. Since the dataset is part of the validation set, some networks could be biased towards having greater performance than the real one (with a test dataset which they have not ever seen) because they were trained on the COCO dataset. The experiments were run on an Nvidia GeForce GTX 1080 GPU.

The selected networks are an implementation of SSD Inception v2, a Faster RCNN Resnet 101, YOLOv3 and a Faster RCNN Resnet 50 FPN. The first and second are downloaded from the TensorFlow detection model zoo [45]. It offers a broad variety of pre-trained networks with metrics. For YOLOv3 the configuration and weights were downloaded from the official documentation and the fourth is included in PyTorch vision model zoo [46]. With this set of different networks, the wide variety of frameworks supported is shown in a real experiment, involving in this experiment TensorFlow, PyTorch and the YOLO-OpenCV module.

In Table 1, the results obtained are shown. For SSD Inception v2, YOLOv3 and Faster RCNN Resnet101 networks the mean inference times are close to the ones provided by the original researchers, slightly higher for the experiments conducted with Detection Metrics. This is probably due to the different GPU used and computational load at the time of the experiment on the computer. TensorFlow’s pre-trained networks and YOLOv3 official results were obtained using an Nvidia GeForce GTX TITAN X card. Regarding the Faster RCNN Resnet50 FPN network the difference is significant, maybe because this PyTorch’s pre-trained network was tested by its authors using 8 V100 GPUs and in the experiment with Detection Metrics a single one GPU was used.

Detection Metrics considers both AP and AR in the evaluation, providing these metrics from a IoU of 0.5 to 0.95 and the mean of each metric for that range. The mAP measured values are also approximately equal to those published by the original researchers, with slightly better numbers when using Detection Metrics in general. The numbers confirm the results provided by the authors of each network. Regarding the network comparison, YOLOv3 is the best-performing network in mAP, as expected.

With this experiment, the use of Detection Metrics for the validation of the results of widely used detection network models and their cross-framework comparison has been illustrated.

### 4.2. Usage on a Deep Learning Real Research Application

This section presents the use of Detection Metrics in the development of a real application, Smart-Traffic-Sensor [47]), proving its usefulness and the added value it provides. This real application monitors road traffic using computer vision. It continuously receives road images as input and generates as output the images with the detected objects, their classification, their predictions and some traffic statistics (see Figure 5).

For this experiment, four neural network architectures were trained using different frameworks (Tensorflow, Darknet and Keras) in the task of detecting vehicles in different lightning and camera conditions. Their performance was compared using Detection Metrics software, especially the automatic Headless evaluation, following a similar strategy as displayed in Figure 4 but with a different set of networks and dataset. These architectures are SSD with a VGG-16 backbone [48], SSD with MobileNet v2 [49] backbone, YOLOv3 [50] and YOLOv4 [41]. For both YOLOv3 and YOLOv4 models, pretrained instances were first used, prior to the retrain with the custom dataset. The models were generated using different deep learning frameworks, showing that Detection Metrics supports all of them. A new dataset with 9774 real traffic images was also created using Detection Metrics [47].

In Table 2, the comparison of the used networks, obtained using the automatic Headless evaluation, is shown. The metrics used to compare the performances are mAP, mAR and the mean inference time in milliseconds. A value of 0.5 is considered as the IoU threshold to consider a prediction as having enough quality to be considered correct. In each row, a different neural network is considered and in the columns, the output statistics are displayed. The experiments were conducted on a computer with a GeForce RTX 3070 GPU.

The results show that the best performing model is the YOLOv4 with transfer learning using the real traffic dataset, which is used in the final Smart-Traffic-Sensor application. The results for YOLOv3 are close to those for YOLOv4.

In another experiment, Detection Metrics toolkit was also used to compare the performance of Smart-Traffic-Sensor and Traffic-Monitor [51], a previous version of the application that did not use deep learning techniques at al. Instead, Traffic-Monitor was based on Support Vector Machines (SVM). The traffic dataset was divided into three different groups, as illustrated in Figure 6: the first one contains images whose conditions are easy to detect (good lightning conditions and high camera resolution), and the second one includes images where the weather conditions are bad and the third one low-quality images. This dataset was the input for Smart-Traffic-Sensor and Traffic-Monitor. Table 3 shows and compares the measured performance computed with Detection Metrics. In the comparison, two versions of Smart-Traffic-Sensor are considered, one using YOLOv3 and another using YOLOv4. The best performance for both mAP and mAR metrics is obtained by Smart-Traffic-Sensor with YOLOv4 for every group of images, obtaining an important improvement from the previous version based on classical machine learning techniques. The enhancement is even greater in bad weather images.

## 5. Conclusions

Detection Metrics, an open-source scientific software for the automatic evaluation of deep learning object detection models, has been presented in this paper and it is its main contribution. We have described its two main workflows for working with object detection networks and large datasets. First, Headless evaluation (Section 3.2), which automatically runs several models independently across large image datasets returning predicted object and experimental objective metrics. Second, Live detection visualization (Section 3.3), which allows users to see the predicted detections on the screen on real time. We have proven how the described workflows and the extra tools available in the toolkit can be used by researchers to develop object detection applications and quantitative objective deep learning model comparisons.

Two experiments have been conducted. The first one replicates the published results from original authors of widely used state-of-the-art detection networks. The second one was the development of a real-world application using DetectionMetrics to create a domain-specific dataset of annotated images and to provide feedback about different detection networks to be embedded in the final application. In this experiment, the application helped in the decision of the best solution for the particular requirements of the research project. With them, we have demonstrated some of the toolkit’s possible use cases and its usefulness in the decision process, saving time for the researchers.

Additionally, the toolkit is open-source, its code can be audited, modified, or extended for different needs of particular scenarios and other use cases, due to its modularity. Its source code is available at https://github.com/JdeRobot/DetectionMetrics, accessed on 27 April 2022.

## Figures and Tables

**Figure 1 sensors-22-04575-f001:**
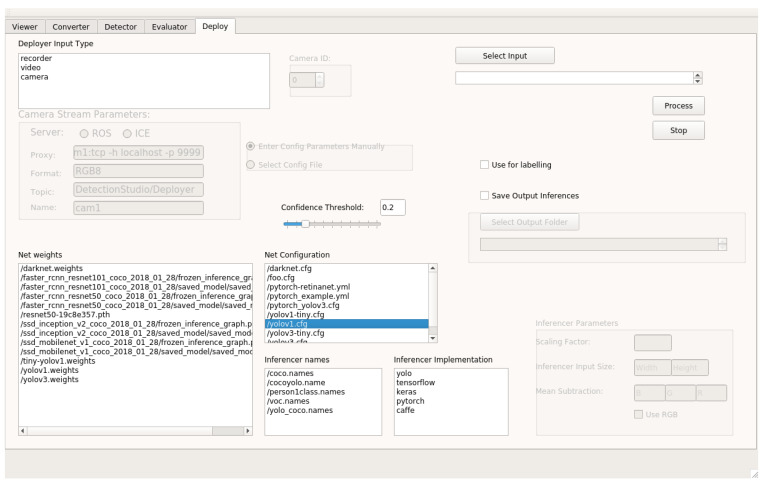
Detection Metrics GUI. The user can select the tool to use from the tool kit and enter the parameters directly using the graphical interface. In addition to the GUI, the headless mode is also available using the command-line and a configuration file to access the functionality.

**Figure 2 sensors-22-04575-f002:**
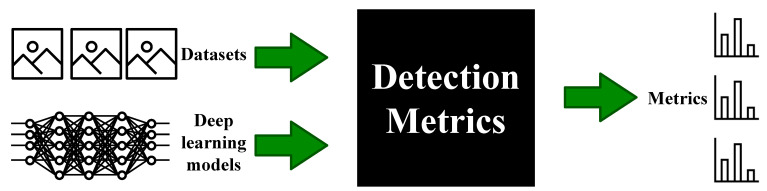
Detection Metrics illustrated as a black box diagram. Detection Metrics receives a batch of datasets and deep learning models as input, calculates all the metrics from combining the datasets and deep learning models and finally outputs the metrics results.

**Figure 3 sensors-22-04575-f003:**
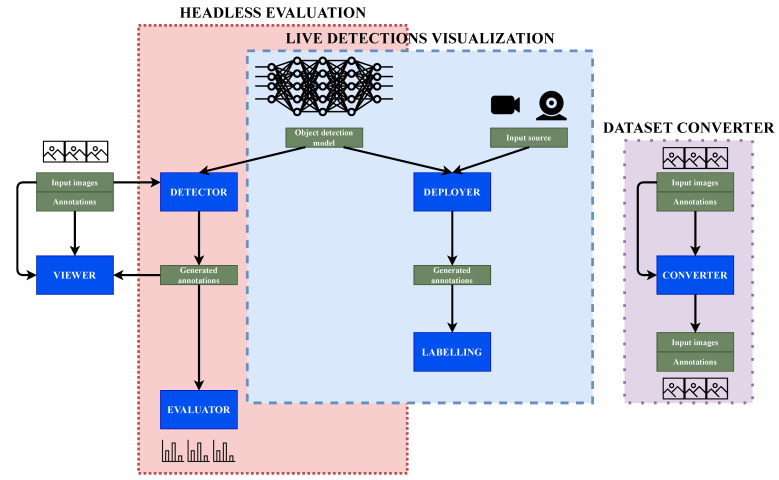
General Detection Metrics architecture. The software provides three main use cases: headless evaluation, live detections visualization and dataset converter. Each of them has a set of tools (in blue), that can be used individually or combined.

**Figure 4 sensors-22-04575-f004:**
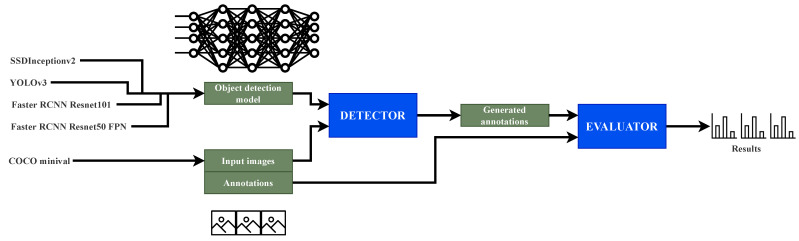
Experiment pipeline using headless evaluation. Detection Metrics receives a set of deep learning models and a dataset and generates annotations with Detector that are the input to Evaluator for obtaining the experimental results.

**Figure 5 sensors-22-04575-f005:**
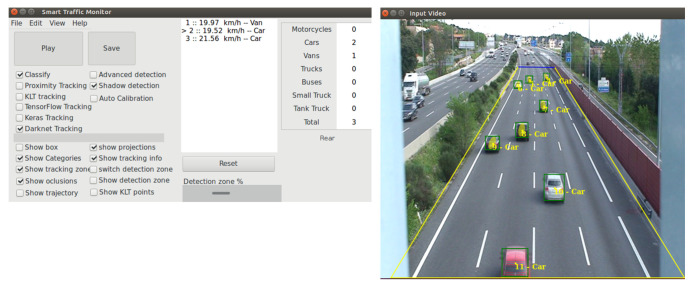
Smart-Traffic-Sensor application has two windows. The window where the options are selected via mainly check-boxes and results are shown as text and the window where the results are shown graphically while the application is running. In the second window, the selected video is played and the detected cars are displayed with a bounding box and their identification.

**Figure 6 sensors-22-04575-f006:**
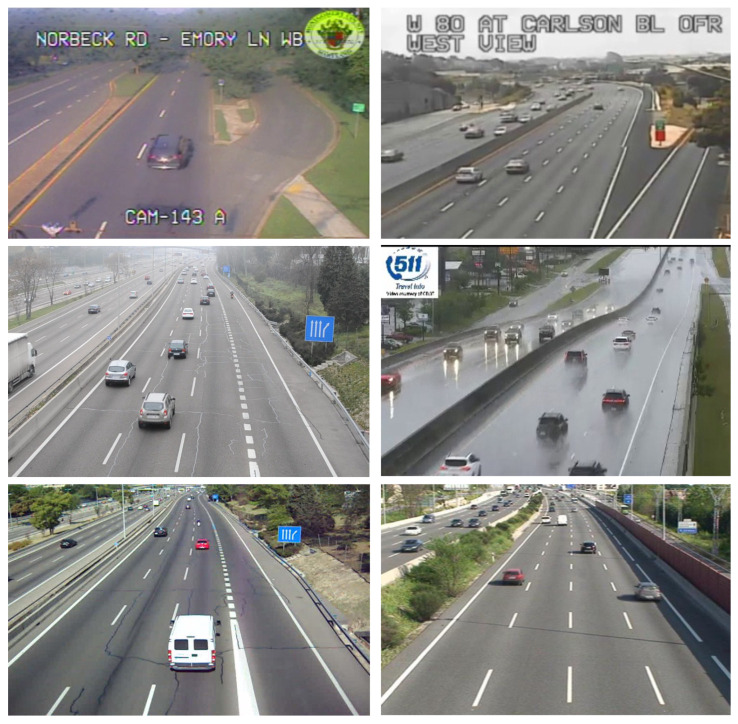
The dataset used for developing *Smart-Traffic-Sensor* includes different image types: low quality, bad weather and good conditions (good lightning conditions and good camera resolution).

**Table 1 sensors-22-04575-t001:** Comparison of official network results with results generated using Detection Metrics. Our software is used to replicate official results of common network architectures programmed in different deep learning frameworks, probing the software capabilities for working with different frameworks and providing common metrics that match the official results. The result *x* is used when the official results do not give that information.

Network	Framework	Published mAP	mAP Using Detection Metrics	Published mAR	mAR Using Detection Metrics	Published Mean Inference Time	Mean Inference Time Using Detection Metrics
SSD Inceptionv2	TensorFlow-Keras	0.24	0.27	x	0.31	42	44
YOLOv3	Darknet	0.55 (IoU = 0.5)	0.47 (IoU = 0.5)	x	0.5 (IoU = 0.5)	29	31
Faster RCNN Resnet101	TensorFlow-Keras	0.32	0.37	x	0.43	106	122
Faster RCNN Resnet50 FPN	PyTorch	0.35	0.37	x	0.46	59	102

**Table 2 sensors-22-04575-t002:** Results of the comparison of networks extracted using Detection Metrics. The results from this experiment were extracted using the software presented in this work. Combining different deep learning frameworks and network architectures, Detection Metrics generates results based on common metrics that provide information for comparing the different strategies easily.

Network	Framework	Transfer Learning	mAP	mAR	Mean Inference Time (ms)
SSD VGG-16	Keras	Yes, retrained	0.7478	0.7831	13
SSD MobileNet v2	TensorFlow	Yes, retrained	0.5484	0.6136	10
YOLOv3	Darknet	No, pretrained	0.4577	0.5843	34
YOLOv3	Darknet	Yes, retrained	0.8926	0.9009	15
YOLOv4	Darknet	No, pretrained	0.4799	0.5879	24
YOLOv4	Darknet	Yes, retrained	0.9056	0.9670	13

**Table 3 sensors-22-04575-t003:** Comparison of systems in different types of dataset images using Detection Metrics. The toolkit provides common detection statistics not only for deep learning systems but also for systems of different nature (*Traffic-Monitor*).

Dataset Image Type	Good Conditions		Bad Weather		Poor Quality	
System Type	mAP	mAR	mAP	mAR	mAP	mAR
Traffic-Monitor	0.4374	0.5940	0.2407	0.3162	0.4479	0.6303
Smart-Traffic-Sensor YOLOv3	0.8926	0.9009	0.9899	0.9926	0.9439	0.9444
Smart-Traffic-Sensor YOLOv4	0.9056	0.9670	0.9904	0.9949	0.9902	0.9911

## Data Availability

Not applicable.

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
