# Peer review of "Open Source Assessment of Deep Learning Visual Object Detection"

_sensors, 2022, doi:10.3390/s22124575_

Round 1

Reviewer 1 Report

This paper introduces an open-source software tool for the assessment of deep learning object detection methods, entitled “Open Source Assessment of Deep Learning Visual Object Detection”. Two main workflows, i.e., headless evaluation and detection visualization, are discussed in the work.

It would be better if the authors consider the comment mentioned below and further improve the manuscript before submitting the final version. The contributions of the work can be presented more clearly. More state-of-the-art studies should be discussed.

Author Response

Thank you for your kind suggestions on the manuscript. We appreciate the comments and review for improving the quality of the article.

Introduction section has been extended with a broader state of the art presentation, including labeling software (line 41), new learning techniques such as few-shot learning (lines 43-46) and other computer vision methodologies (lines 46-48), and the state of the art section, more detailed explanations have been introduced and equations for components that needed them (Performance metrics subsection). The references you suggested have been included.

Regarding the improvement of the methods, we have extended the description of the software tool, with a more detailed explanation of its features on dataset generation and automation of the experimental process.We have updated a figure explaining the pipeline structure of experiment 4.1.

Discussion has been moved to Experimental results (renamed Experimental results and discussion) and that section has been renamed to Conclusions, updating its contexts and pointing to the main contributions of our manuscript, especially Detection Metrics software tool.

We have not found any further comment in your review regarding the "comment mentioned below". 

Reviewer 2 Report

This paper exploits and assess the performance of open source deep learning visual object detection methods. In general, this paper is interesting, and should consider the following comments:

  1. The discussion section is too short and lack of insightful opinions.
  2. There is no conclusion section, which makes this paper incomplete.
  3. The figures in this paper are too large and some of them are not important or necessary at all. 
  4. There is no visualization of experimental results, which makes it less readable.
  5. The metrics used in this paper are not adequate to compare performance between different models. 
  6. The tables used in this paper need some revisions to achieve better spacing.
  7. There are no equations to explain some key parameters such as IOU, AP and AR.
  8. This paper simply employs some existing models without obvious innovation.
  9. Other object detection approaches should be reviewed in the introduction section, e.g.,  A hybrid active contour model based on pre-fitting energy and adaptive functions for fast image segmentation, Pattern Recognition Letters; A level set method based on additive bias correction for image segmentation, Expert Systems with Applications.

Author Response

Thank you for your kind suggestions on the manuscript. We appreciate the comments and review for improving the quality of the article.

We have reviewed the introduction section and extended it with a broader presentation of computer vision state-of-the-art projects, as suggested [your items 1], including labeling software (line 41), new learning techniques such as few-shot learning (lines 43-46) and other computer vision methodologies (lines 46-48). The references you suggested have been included [your item 9].

In the state of the art section, more detailed explanations have been introduced and equations for components that needed them (Performance metrics subsection). 

Regarding the improvement of the methods, we have extended the description of the software tool, with a more detailed explanation of its features on dataset generation and automation of the experimental process.We have updated a figure explaining the pipeline structure of experiment 4.1, replacing  another figure of minor relevance [your item 3].

Discussion has been moved to Experimental results (renamed Experimental results and discussion) and that section has been renamed to Conclusions, updating its contexts and pointing to the main contributions of our manuscript, especially Detection Metrics software tool [your item 2].  This scientific tool itself is the main innovation of the paper [your item 8].

We thank the reviewer for raising a concern about the metrics used in the manuscript. After reviewing the literature, benchmarks, and international competitions, we recognize that the metrics used are fair and common across the field [your item 5]. As an example, part of the experimental result from experiment 4.2 (Table 1)  are extracted from other researchers' results and we use them as a baseline for comparison against our experimental results.

The equations to explain IOU, AP and AR have been included [your item 7].

We agree with the comment regarding the tables and have updated the formatting of them to a more readable format making the result presentation easier to follow [your items 4, 6]. The tables shown at both experiments (4.1 and 4.2) and Figure 5 are the experimental results. 

Reviewer 3 Report

The manuscript introduces an open source solution for the assessment of object detection based on different deep learning algorithms. The solution supports several datasets and learning frameworks, and provide the evaluation by specific performance metrics accordingly. The experiments are also reported to present the usability of the proposed solution. Finally, the solution is open to the public through GitHub.

Overall, the manuscript is well-organized. However, it is suggested that the design and the contribution of the proposed solution can be elaborated in more detail. Otherwise, readers, engineers, or researchers might refer to other review studies and choose appropriate learning frameworks according to their requirements. Detail suggestions are stated as the following.

1. The design of the proposed solution can be elaborated in more detail. As depicted in Figure 3, the detector generates detection results based on different object detection models. The underlying design or major components of the detector can be introduced. For instance, the interface design between detector and different object detection models might be an important component. In addition, the extensibility should be considered because new object detection models will be developed and evaluated through the proposed solution. Finally, from the perspective of users, how to leverage new data sets or new detection models can be elaborated.

2. The evaluation process performed by the proposed solution can be elaborated in more detail. Is it fully automatic? In addition, Is there any setup or teardown process performed by the solution, especially in the performance evaluation? For instance, a reboot can be performed after the evaluation to make sure the resources can be released for the next evaluation. Thus, the results of the comparison and the performance evaluation are more convincing.

3. The benefit of the proposed solution can be elaborated in more detail. In other words, the insight or benefit brought by the proposed solution should be more than those by the review studies. Can readers get more (hidden but important) information? Can engineers make appropriate decisions based on their particular requirements? Can engineers save time on evaluation? It is suggested that the authors can provide more information for readers.

Author Response

Thank you for your kind suggestions on the manuscript. We appreciate the comments and review for improving the quality of the article.

We have reviewed the introduction section and extended it with a broader presentation of computer vision state-of-the-art projects, including labeling software (line 41), new learning techniques such as few-shot learning (lines 43-46) and other computer vision methodologies (lines 46-48).

In the state of the art section, more detailed explanations have been introduced and equations for components that needed them (Performance metrics subsection).

We have updated a figure explaining the pipeline structure of experiment 4.1, replacing another figure of minor relevance. We have updated the formatting of the tables to a more readable format making the result presentation easier to follow.

[item 1]

We have extended the description of the proposed solution, especially discussing that the Detector tool has a wrapper for each framework, making it easy for extension in case a new framework is available ('In order to provide the different frameworks support, Detector has interfaces for each of them, connecting the actual deep learning framework to the tool in an agnostic way that prevents the user from facing any complexity. Thanks to the modularity of the Detector tool and the fact that the project is open-source, when new deep learning frameworks are created, they can be added without modifying the inner structure of Detector.'). The tool is extensible.

We have also introduced more details regarding the dataset generation and the Labelling tool ('This workflow can be interesting for generating completely new datasets for research of industrial purposes, creating first the proposals that some object detection model predicts and then adjusting the result by modifying the predicted bounding boxes and predicted classes or adding new predictions that are relevant for a specific task.').

[item 2]

The software evaluation is automatic and we apologize for not making it clear previously. We have modified the general manuscript introducing it, making it clear that several experiments can be conducted sequentially without detriment to the performance of the software ('A researcher can determine a set of experiments that will run independently and unattended (fully autonomous), retrieving the final experiment report with the objective metrics that will help detect flaws and advantages for each model in each scenario.').

[item 3]

Discussion has been moved to Experimental results (renamed Experimental results and discussion) and that section has been renamed to Conclusions, updating its contexts and pointing to the main contributions of our manuscript, especially Detection Metrics software tool. In the Conclusion section, we have also added details on how the decision-making process can be improved using the software, saving researchers time.

Experiment 4.2 is an illustrative example of how engineers may make appropriate decisions based on the tool. DetectionMetrics allows the comparison of different network models or even different ways to train them in order to successfully meet the particular requirements of an application.

Round 2

Reviewer 2 Report

The authors have addressed all my comments.

Author Response

Thanks for your comments.

Reviewer 3 Report

The refined manuscript addresses the comments and suggestions appropriately. Some minor suggestions are listed as the following.

1. Please check formula (2) in page 3. It should be Recall.

2. Please give the description to the "ROS Node" in line 211.

3. In Table 1, the published mean inference time of Faster RCNN ResNet50 FPN is 59. The mean inference time using Detection Metrics proposed in the manuscript of the same network is 102. The difference is significant compared to other networks. If the goal of the proposed Detection Metrics is to be an evaluation standard, the difference should be analyzed carefully. 

4. Please check the numbers of mAR and Mean inference time (ms) of YOLOv4 (pretrained and retrained).

Author Response

We would like to thank you again for your kind suggestions and comments on our draft manuscript Open Source Assessment of Deep Learning Visual Object Detection and for the opportunity to submit a revised version of the document. We appreciate the time and effort dedicated to the review of the manuscript for improving its quality.

After reviewing the comments and suggestions, we have generated an updated document, including all the suggested improvements:

1.- We have updated Formula (2) on page 3 since there was a misspelled word that should be Recall, as you pointed out.

2.- We have extended the description of the ROS Node use case (lines 208-216), connecting it to the Robot Operating System (ROS) and presenting an example of a possible use.

3.- We have revisited the results in Table 1. There was an error in the last row in column Framework, instead of Keras, the framework is PyTorch. We have made sure that the results in the table are correct and we have extended the explanation in lines 392-400, clarifying that the PyTorch pre-trained model was tested using 8 GPUs instead of 1 when using Detection Metrics. The number of GPUs and their versions explain the difference between our experimental results and those from the onetwork authors.

4.- We reviewed the results in Table 2, especially YOLOv4 numbers, and updated them accordingly, since there was an error on mAR and Mean inference time and the numbers were not correct.